# Inner Ear Diagnostics and Drug Delivery via Microneedles

**DOI:** 10.3390/jcm11185474

**Published:** 2022-09-17

**Authors:** Stephen Leong, Aykut Aksit, Sharon J. Feng, Jeffrey W. Kysar, Anil K. Lalwani

**Affiliations:** 1Vagelos College of Physicians & Surgeons, Columbia University Irving Medical Center, New York, NY 10032, USA; 2Department of Mechanical Engineering, Columbia University, New York, NY 10027, USA; 3Department of Otolaryngology—Head & Neck Surgery, New-York Presbyterian/Columbia University Irving Medical Center, New York, NY 10032, USA

**Keywords:** microneedle, round window membrane, intracochlear delivery, precision medicine, gene therapy

## Abstract

Objectives: Precision medicine for inner ear disorders has seen significant advances in recent years. However, unreliable access to the inner ear has impeded diagnostics and therapeutic delivery. The purpose of this review is to describe the development, production, and utility of novel microneedles for intracochlear access. Methods: We summarize the current work on microneedles developed using two-photon polymerization (2PP) lithography for perforation of the round window membrane (RWM). We contextualize our findings with the existing literature in intracochlear diagnostics and delivery. Results: Two-photon polymerization lithography produces microneedles capable of perforating human and guinea pig RWMs without structural or functional damage. Solid microneedles may be used to perforate guinea pig RWMs in vivo with full reconstitution of the membrane in 48–72 h, and hollow microneedles may be used to aspirate perilymph or inject therapeutics into the inner ear. Microneedles produced with two-photon templated electrodeposition (2PTE) have greater strength and biocompatibility and may be used to perforate human RWMs. Conclusions: Microneedles produced with 2PP lithography and 2PTE can safely and reliably perforate the RWM for intracochlear access. This technology is groundbreaking and enabling in the field of inner ear precision medicine.

## 1. Introduction

The ears are vital to one’s perception of and interaction with the world, providing constant information to the brain to allow for both effective hearing and balance. Inner ear dysfunction—resulting from a combination of genetic and environmental factors, and characterized by hearing loss, tinnitus, and vertigo—is quite prevalent in the general population (Table 1) [1]. Untreated, auditory and vestibular disturbance can significantly impact function and can have debilitating effects on one’s quality of life. With the identification of over a hundred deafness genes and the recent demonstration of mammalian hair cell regeneration, stem cell therapy, and gene editing technology, we are on the cusp of precision therapy for inner ear disorders [2].

A significant impediment to implementing precision medicine for inner ear disorders is safe and reliable access to the inner ear for diagnostics and therapeutic delivery. Without a means to sample inner ear fluid for electrochemical, RNA, or proteomic analysis, precise intervention is not possible. Furthermore, current options for intracochlear delivery, including systemic administration, intratympanic (IT) injection, and direct injection into the cochlea, are imprecise. Although systemic administration may achieve high drug levels in the cochlea, it is more frequently associated with systemic toxicity. IT injection is a more precise delivery method, but is hampered by variable efficacy, as it must rely on simple diffusion across the round window membrane (RWM). Additionally, the injected medication can leak down the Eustachian tube, be impeded by debris in the round window niche, or escape out of the external canal, thus resulting in highly variable medication levels between patients [3]. Direct placement of therapeutic agents on the RWM in a biodegradable carrier substance, such as gelatin, hydrogel, or nanoparticles, may overcome some of these limitations [4,5,6]; however, the rate of drug delivery to the inner ear is inevitably limited by molecular diffusion across the RWM. These limitations may have been responsible for the recent failures of two promising large clinical trials: AM-111 in the treatment of sudden sensorineural hearing loss, and sustained release dexamethasone in Poloxamer 407 gel for Meniere’s Disease [7,8].

Direct intracochlear drug administration results in significantly higher and less variable drug levels compared to IT injection, with a much smaller concentration gradient from base to apex [9]. Various methods of intracochlear delivery have been developed, including osmotic mini-pump infusion into the scala tympani via the round window [10,11,12,13], infusion or microinjection into the scala tympani through a cochleostomy [13,14,15,16,17], and intracochlear injection through the RWM [18,19,20,21]. Though these methods are a step toward precise inner ear delivery, all of them breach the inner ear and consequently risk hearing impairment. A safe and reliable method for intracochlear delivery thus remains to be developed.

The RWM is the only soft tissue portal from the middle ear into the cochlea and, therefore, is an ideal candidate for intracochlear access. Microneedle-mediated perforation of the RWM is a novel means of achieving intracochlear access and can facilitate reliable and predictable perforation of the RWM, with minimal anatomic and functional damage [22,23]. Using microneedles, drug concentrations within the inner ear may be controlled with a precision that IT injections cannot provide. The application of microneedles is not a departure from current clinical practices, but rather a natural progression from the current practice. Microneedle technology promises increased safety and efficacy over the current techniques for inner ear therapy, and may potentially be applied in an office setting.

## 2. Properties of the Round Window Membrane

A significant challenge to intracochlear delivery is perforation of the RWM without inducing tearing or ripping. As will be discussed in a later section, the size of such a perforation must necessarily be at least an order of magnitude smaller than the RWM itself, which means that the perforations should be no wider than about 200 µm through the 2 mm wide and 70–80 µm thick RWM. Designing microneedles for this purpose requires knowledge of the RWM microanatomy and mechanical properties. The RWM has a connective tissue core containing fibroblasts, collagen, and elastic fibers, providing mechanical strength to the RWM to bear the perilymphatic pressure [24,25]. Over the last several years, the microanatomy and mechanical properties of the RWM have been extensively studied, and a detailed finite element model (FEM) simulation that has directly impacted microneedle design has been developed [26,27,28,29].

Using micro-CT (µ-CT) with 1 µm resolution and white laser interferometry, we demonstrated in a guinea pig model that much of the surface of the RWM can be approximated as a hyperbolic paraboloid (HP)—like a saddle or a Pringle potato chip—that sits in a bony sulcus reminiscent of the tympanic annulus and the tympanic membrane. Using immunostaining and sectioning, confocal and multi-photon microscopy, and scanning electron microscopy (SEM), we showed that RWM fibers are highly organized and share directionality and dispersion characteristics. In large portions of the RWM, fibers are primarily oriented in the direction of zero curvature which allows them to remain as straight as possible in their physiologically natural configuration. There is a strong correlation between elastic and collagen fiber directionality and the distance along the axis of the cochlea. The fibers also follow a direction that improves the behavior of the RWM when it is subjected to increased perilymphatic pressure from the inner ear side, which is consistent with the fact that the perilymph of a guinea pig (GP) is usually at a nominal physiological internal pressure of 0.2 kPa.

## 3. Microneedle Design and Testing

Since they were first demonstrated in 1998, microneedles have been extensively researched for sampling of biological fluids and therapeutic delivery [30]. While early research focused on transdermal sampling and delivery, microneedles have subsequently been used in many different tissues, including the oral cavity, genitourinary tract, gastrointestinal tract, vascular wall, eye, and skin [31,32,33,34]. Our group has specifically focused on the development of microneedle technology for inner ear access [35,36,37,38].

### 3.1. Design for RWM Perforation

A prototypical solid microneedle design from our group is shown in Figure 1a. It starts from a tip of radius $R_{t}$ and tapers at an angle α to a constant shaft diameter $D_{n}$, with a taper-plus-shaft length of L and a base with maximum diameter $D_{b}$ designed to fit into the lumen of a blunt stainless-steel needle. A Luer lock is affixed to the other end of the stainless-steel needle.

The human RWM is about 2 mm in diameter and 70–80 µm thick, with collagen and elastic fibers that endow the RWM with stiffness, strength and toughness; furthermore, the RWM is under a tensile prestrain [40]. Microneedles are designed to create perforations through the RWM to deliver drugs and aspirate fluids for diagnosis. The perforation must be minimally traumatic to promote rapid healing and avoid anatomic and functional consequences. Thus, the tip radius $R_{t}$ should be sufficiently small to penetrate the RWM, while cutting as few collagen and elastic fibers as possible. Fortuitously, such a process minimizes the peak perforation force $F_{p}$ that the microneedle exerts on the RWM and also minimizes the pressure increase in the cochlea during perforation.

Our group has found that the two most important design criteria are: (1) an ultra-sharp tip $R_{t}$ to minimize both trauma and perforation force; (2) high strength and ductility to ensure the microneedle has a safe “bend not break” failure mode. Once these two criteria are fulfilled, the low perforation force $F_{p}$, coupled with a strong and ductile material, allow length L, shaft diameter $D_{n}$, and angle α to span large ranges while still maintaining microneedle structural integrity with a large factor of safety. We can then specify L, $D_{n}$, and α based on medical needs, rather than compromising these values for the sake of structural integrity.

Nonetheless, fabrication of strong, ductile, ultra-sharp microneedles is very challenging, and even more so for microneedles that are hollow or have complex geometries. To overcome this challenge, our group has developed a new paradigm for fabrication of microneedles.

### 3.2. Microneedle Fabrication

Methods to manufacture microneedles to perforate the RWM require three attributes: (1) accuracy and precision leading to ultra-sharp needles; (2) strong and ductile material; and (3) high design freedom to fabricate complex needle geometries for middle and inner ear anatomy. Figure 2 depicts these in a Venn diagram.

Existing microneedle manufacturing methods include: micromachining, direct writing techniques, laser machining, micromilling, electric discharge machining, laser sintering, electroplating and various combinations of lithography, molding, and hot/soft embossing techniques [30,41,42]. Materials used include: silicon, polysilicon, steel, nickel, gold, titanium, and a variety of different polymers. None of these methods and materials encompass all three attributes in Figure 2.

Our group has therefore employed a new, enabling manufacturing technology called two-photon polymerization (2PP) lithography that has become commercially available in the past few years [43]. The 2PP lithography method is an additive manufacturing (i.e., 3D printing) process that can produce highly complex geometries out of hard polymers with voxel spatial resolution approaching 100 nm. The sub-micrometer precision and accuracy, combined with the design freedom of 3D printing, enable the direct 3D writing of polymeric microneedles. Figure 1a shows a polymeric needle from our group with tip radius $R_{t}=0.5$ µm and a shaft diameter $D_{n}=$100 µm printed with 2PP lithography and mounted on the end of a 23-gauge blunt hollow stainless-steel needle [35].

To assess the effectiveness of the polymeric microneedles, our group designed—based on the RWM anatomic and mechanical characteristics described above—microneedles with $D_{n}=100$ µm, α=18°, and $L_{n}=200$ µm, and performed in vitro studies of GP RWM perforation [35]. Figure 3a shows a confocal microscopy image of the perforation introduced at the center of a GP RWM with a mean $F_{p}=1.2$ mN; Figure 3b shows collagen and connective fiber separation, and demonstrates that the length of the lens-shaped perforation is approximately the same as the microneedle diameter.

### 3.3. Anatomical and Functional Consequences of Perforation

To assess anatomic and functional consequences of microneedles on the RWM, our group used the same microneedles to perforate the GP RWM in vivo [36]. The ultra-sharp microneedles created precise, accurate, and stable perforations with separation of connective fibers (Figure 4). Confocal microscopy showed that the RWM perforation began to heal by 24 h and completely healed by 1 week; subsequently, we showed that complete closure of the RWM occurred between 48 and 72 h. Perforations could not be detected histologically at 1 week. From audiometric measurements, including compound action potential (CAP) and distortion product otoacoustic emissions (DPOAE) at 0–2 h, 24 h, 48 h, and 1-week post-perforation, there were no measurable audiologic consequences, although covert hearing loss cannot be ruled out. Of note, these experiments were performed in healthy GPs; models of auditory pathology, especially those producing pressure abnormalities in the inner ear, may result in different RWM healing properties.

Based on the mechanical properties of the human RWM, the human microneedle design was modified from the GP microneedle design to account for the stronger and thicker human RWM, with α=60°, $D_{n}=150$ µm, and L=480 µm; this microneedle is shown in Figure 1c. The polymeric microneedles designed for human use created precise and stable perforations that were slit-shaped via fiber separation, with a distinct major axis equal to the microneedle diameter, and aligned with the predominant fiber direction; the microneedles maintained their integrity during perforation [37]. Approximately $F_{p}=60$ mN was required to perforate the RWM and microneedles needed to be displaced inward approximately 300 μm, which is sufficiently small so that the microneedles do not touch the closest structures behind the RWM during perforation, thus avoiding cochlear trauma. In summary, these microneedles designed for perforating the human RWM were durable, created precise perforations, and avoided cochlear trauma.

Thus, a microneedle with an ultra-sharp tip radius $R_{t}\approx 1$ µm is an achievable design specification with 2PP lithography. Furthermore, stable perforations of 100 µm in a GP model and 150 µm in human tissue can be introduced safely. Perforations of this size scale are known to significantly enhance the rate of diffusion of molecules across the GP RWM [44].

### 3.4. Design Freedom in Microneedle Synthesis

The 2PP method provides impressive flexibility to create polymeric microneedles with complex geometries. Figure 1f shows an array of five microneedles—each 100 µm in diameter—secured to a common base that is mounted on a stainless-steel blunt needle. Figure 1e shows five “crown” needles with different diameters for creating large perforations on the RWM, through which a cochlear implant can be inserted.

While the polymeric material from 2PP has good strength and ductility, it is not as strong and ductile as a metal, nor is the specific polymeric material biocompatible. Therefore, our group developed a new method to fabricate ultra-sharp metallic microneedles using a technique known as two-photon templated electrodeposition (2PTE). First, 2PP lithography is used to “print” polymeric mold structures containing cavities in the shape of the desired needle. The cavities are then filled with copper via electrochemical deposition. The polymeric molds are then dissolved, and the needles are recovered and mounted on a blunt stainless-steel needle, as shown in Figure 1b [38]. A biocompatible metallic coating is also applied onto the microneedles, consisting of a 1.5 µm conformal film of nickel followed by a 30–100 nm conformal film of gold. The final microneedle has a tip radius of 1.5 µm and shaft diameter of 100 µm, and has been tested in vitro in a GP RWM. The mean peak perforation was 4 mN and the ultra-sharp tip successfully separated RWM fibers.

The 2PTE process provides sub-micrometer resolution to create precise, ultra-sharp, high-ductility, high-strength, and biocompatible metallic microneedles that have significant design freedom [38]. Figure 1g illustrates one of the complex geometries that can be fabricated via 2PTE; specifically, a 410 µm crown needle to facilitate cochlear implantation across the RWM is shown. The combination of design freedom, precision, and material choice of the 2PTE process is unique in microneedle manufacturing, is ideal for microneedle development for inner ear precision medicine, and encompasses all three attributes outlined in Figure 2.

## 4. Clinical Applications of Microneedles

Our microneedle technology is critical for the field of otology because it allows for direct access into the cochlea without lasting effects on hearing. Thus, our microneedle technology makes precision medicine of the inner ear possible, in both diagnostic and therapeutic realms. We have developed hollow microneedles that are capable of both aspiration and injection of fluid; in this section, the diagnostic utility of perilymph aspiration and the therapeutic utility of direct intracochlear injection is discussed.

### 4.1. Hollow Microneedles for Aspiration

To assess the potential of aspiration for the diagnosis of cochlear disorders, our group designed and 2PP printed hollow microneedles with $D_{n}=100$ µm, L=435 µm, α=24°, and 35 µm diameter lumen and mounted them atop a 30-gauge blunt hollow stainless-steel needle, as shown in Figure 1d [39]. We then aspirated 1 µL of GP perilymph across the RWM in vivo for proteomic analysis. Over 400 proteins were identified; the inner ear protein cochlin, widely recognized as a perilymph marker, as well as proteins from the heat shock protein family, including heat shock protein 70, were detected in all samples tested. Results are shown in Figure 5. There were no measurable shifts in hearing thresholds, and perforations healed completely within 72 h. The ability to collect perilymph will overcome the methodological limitation of prior studies that required dissecting the whole cochlea from animals for adequate tissue samples.

Our group further tested the clinical utility of cochlear aspiration by determining if proteomic differences in systemic versus IT delivery of steroids could be detected in perilymph. Previous studies have demonstrated that local administration results in greater perilymph concentrations and favors the base, whereas systemic administration favors the apex [45,46]. Additionally, intratympanically delivered glucocorticoids have been found to affect thousands more inner ear genes compared to systemically delivered glucocorticoids in mice [47]. Through aspiration of 1 µL of GP perilymph using hollow microneedles, we demonstrated that systemically administered dexamethasone results in greater modulation of perilymph proteins compared to IT dexamethasone, with 14 modulated proteins in the systemic group and 3 modulated proteins in the IT group [48]. In both groups, the growth factor VGF was significantly upregulated and the regulatory protein 14-3-3γ was downregulated; in particular, upregulation of VGF suggests an otoprotective role for steroids administered both systemically and via IT injection. Increased modulation of protein expression with systemically administered steroids conflicts with previous studies [47], but may suggest greater off-target effects for systemic therapy. In summary, our ability to distinguish between systemically and locally administered glucocorticoids via microneedle aspiration of perilymph further supports the efficacy of our technology.

### 4.2. Microneedles for Diffusion and Injection

Microperforations across the RWM can enhance diffusion of therapeutics across the RWM. Our group has shown that a single perforation occupying 0.22% of RWM leads to a 35-fold increase in diffusion of Rhodamine B (chosen because of its similarity to dexamethasone and gentamicin)—consistent with our predictions based on mathematical modeling [44]. Interestingly, the use of Poloxamer-407 gel as a reservoir for the Rhodamine B results in a slower diffusion rate than PBS solution, but with a smaller standard deviation. One concern when creating perforations within the RWM is the potential leakage of perilymph from the scala tympani into the middle ear due to perilymph pressure [19,49]. As the fluidic resistance to flow is inversely proportional to the diameter of the perforation to the fourth power, the risk of leakage can be further reduced by using multiple smaller microperforations in place of a single larger perforation to mediate drug diffusion [50]. We tested diffusion across multiple smaller holes using a filter paper model of the RWM in a horizontal Valia–Chien diffusion cell and a phosphate buffered saline (PBS) fluid in the reservoir donor chamber [51]. Our group demonstrated that the diffusion rate across the membrane from a liquid reservoir was directly correlated to aggregate area of perforation; thus, multiple smaller holes—potentially introduced by an array of microneedles—are equivalent to a single larger perforation in enhancing diffusion but have lower risk of perilymph leakage.

While solid needles have been found extremely successful in enhancing diffusion across the RWM, direct injection of the therapeutic into the cochlea is attractive as a more precise method of delivery. Direct injection has been proposed by several companies, including Akouos and Decibel. Since our 2PTE fabrication technology can create hollow microneedles that can be attached on top of common syringe needles, it is possible to directly inject precise amounts of therapeutics directly across the RWM. In guinea pigs, patency of the cochlear aqueduct allows for direct injection of agents without significant efflux of fluid through the perforation created by the microneedle. However, pressure relief through the cochlear aqueduct likely requires some level of pressure buildup within the cochlea; our group has demonstrated that injection of large volumes of fluid into the GP cochlea through a single-lumen hollow microneedle results in high-frequency hearing loss, which can be entirely avoided with injection of smaller volumes. In humans, the cochlear aqueduct is closed and direct injection will likely result in immediate efflux through the very perforation created. Solutions such as perforation of the stapes, semicircular canal, or apex of cochlea have been proposed to overcome this issue by providing a vent release for escape of cochlear fluid—but are invasive and damaging. As one possible solution, our group has designed a microneedle system with a second lumen that can act as a vent, so that cochlear fluid volume remains constant during injection of a large therapeutic volume (Figure 1h). Our group will continue developing microneedle technology to enable safe injection of therapeutics directly across the RWM.

### 4.3. Office-Based Diagnostic and Therapeutic Intervention

Our group has additionally been developing an endoscopic approach for microneedle-mediated perforation of the RWM and injection of agents. By mounting a hollow microneedle at the tip of a middle ear micro-endoscope, we have successfully perforated human cadaveric RWMs. Further development of this technology will allow our microneedle technology to be translated into the office setting, where otolaryngologists may use microneedles to deliver therapeutics as part of routine outpatient care.

## 5. Conclusions

Many current technologies used to access the inner ear are inherently traumatic and result in significant hearing loss. Microneedles offer a solution to this problem, for both diagnostic and therapeutic purposes. Our microneedle technology allows for direct sampling of perilymph and makes the clinical translation of a variety of intracochlear therapies possible. With our technology, we believe that precision therapeutics such as gene therapy will take hold in the field and revolutionize treatment for hearing loss.

## Figures and Tables

**Figure 1 jcm-11-05474-f001:**
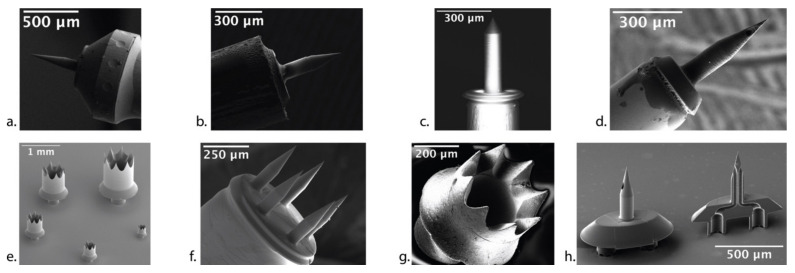
Suite of microneedles developed for the inner ear. (**a**) Solid polymeric microneedle. (**b**) Solid metallic microneedle. (**c**) Solid polymeric microneedle for human RWM use. (**d**) Hollow microneedle for perilymph aspiration and direct intracochlear injection. (**e**) Five differently sized “crown” needles to facilitate cochlear implantation. (**f**) Microneedle array for opening simultaneous microperforations on the RWM. (**g**) A 410 µm sized “crown” needle for cochlear implantation, fabricated via 2PTE. (**h**) Dual-lumen microneedle for simultaneous aspiration and injection of fluids across the RWM. (Adapted with permission from Ref. [35], 2018, Ref. [37], 2020, Ref. [38], 2020, Ref. [39], 2021, Jeffrey W. Kysar, PhD and Anil K. Lalwani, MD).

**Figure 2 jcm-11-05474-f002:**
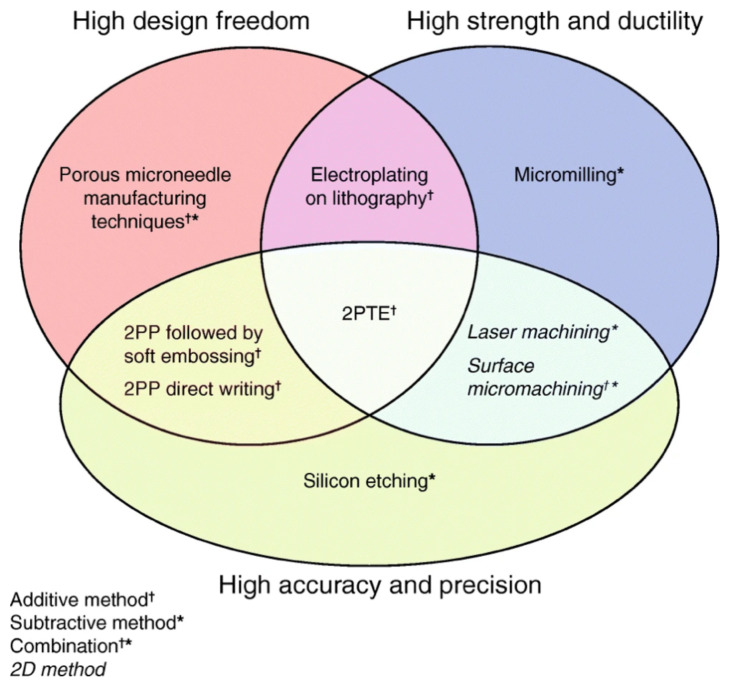
Attributes of different microneedle manufacturing techniques. ^†^ Additive method. * Subtractive method. ^†^* Combination method. Italics: 2D method. (Reprinted with permission from Ref. [38], 2020, Jeffrey W. Kysar, PhD and Anil K. Lalwani, MD).

**Figure 3 jcm-11-05474-f003:**
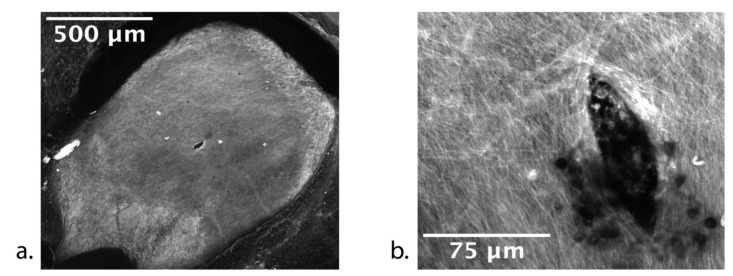
Confocal image of a guinea pig RWM around a perforation with (**a**) low magnification and (**b**) high magnification showing connective fibers of the membrane. (Reprinted with permission from Ref. [35], 2018, Jeffrey W. Kysar, PhD and Anil K. Lalwani, MD).

**Figure 4 jcm-11-05474-f004:**
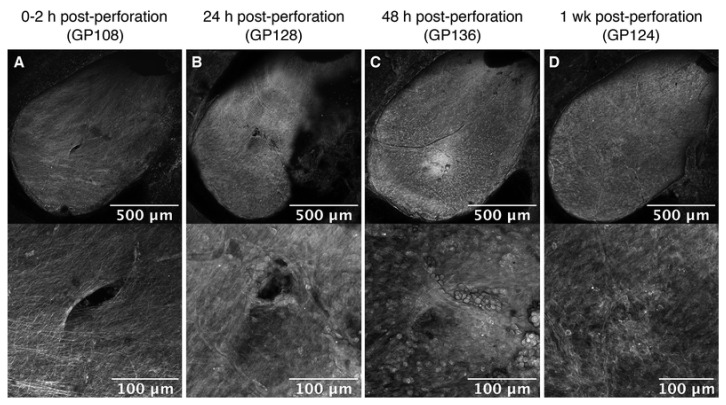
Guinea pig RWM healing after perforation with 100µm-diameter microneedle perforations under low magnification (**top**) and high magnification (**bottom**) at (**A**) 0–2 h, (**B**) 24 h, (**C**) 48 h, and (**D**) 1-week post-perforation. (Reprinted with permission from Ref. [36], 2020, Jeffrey W. Kysar, PhD and Anil K. Lalwani, MD).

**Figure 5 jcm-11-05474-f005:**
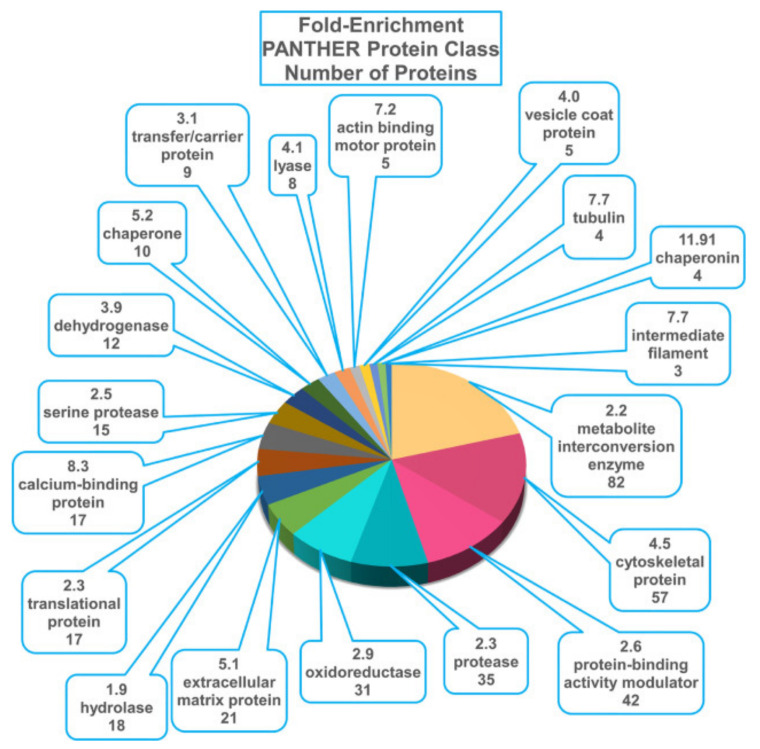
Composition of guinea pig perilymph proteome, by functional categories. The 620 gene names were searched against the mouse gene list in PANTHER (http://www/pantherdb.org (accessed on 1 September 2020)) to determine the distribution of proteins across functional classes. The fold-enrichment, PANTHER protein class, and the number of proteins within each class are presented. (Reprinted with permission from Ref. [39], 2021, Jeffrey W. Kysar, PhD and Anil K. Lalwani, MD).

**Table 1 jcm-11-05474-t001:** Hearing loss data based on National Health Statistics Reports 2020 [1].

	All Ages 18+	Ages 18–44	Ages 45–64	Ages 65–74	Age 75+
Severe hearing loss	2.4% (2.0–2.8)	0.4% (0.2–0.6)	1.9% (1.5–2.5)	6.0% (4.2–8.3)	13.4% (10.2–17.2)
Mild–moderate hearing loss	13.6% (12.7–14.4)	5.3% (4.6–6.1)	16.7% (15.3–18.2)	31.2% (28.0–34.6)	36.4% (31.2–41.9)
Balance issues	18.7% (17.7–19.7)	14.2% (12.9–15.6)	21.1% (19.5–22.7)	27.1% (24.0–30.4)	30.1% (25.3–35.2)

## Data Availability

Not applicable.

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
