# Peer review of "Inner Ear Diagnostics and Drug Delivery via Microneedles"

_jcm, 2022, doi:10.3390/jcm11185474_

Round 1
Reviewer 1 Report
In this paper, the Authors describe the development, production, and utility of novel precision medicine techniques that utilize microneedles across the round window membrane to access the cochlear fluids.
The manuscript is well-written and organized, and it is easy to follow. The Introduction is concise and complete. The presented techniques are original and clearly explained in both text and figure, and they have proven to provide reliable and safe access to the inner ear for diagnostics and therapeutic delivery.
The only concern I have is that in the Introduction the references need to be updated.
Author Response
Comment: The only concern I have is that in the Introduction the references need to be updated.
Response: We agree that many of the references are outdated in the introduction. We have updated our references to include newer sources. Please see the updated bibliography (lines 369-500).
Reviewer 2 Report
This manuscript presents a comprehensive review of an innovative microneedle technology allowing access to the inner ear through the round window.
The whole is clear, the subject being of great interest with the current development of new therapies for cochleo-vestibular pathologies.
My comments therefore concern minor points.
Line 68: the term "elegant" seems too subjective to me and could be replaced by a more neutral term
Line 69: there is a certain paradox in this sentence, since a perforation is necessarily a form of anatomical damage. I think the wording could be less categorical, such as "with minimal anatomic or functional damage".
Line 142- Line 185-Figure 3: The statement that the fibers are spread without being cut seems to me to be difficult to affirm or deny. This is a microscopic observation in which a perforation is observed, the state of the fibers is difficult to analyze. This statement could be nuanced.
Lines 194 to 203: the data show complete healing in one week. I think it should be remembered here that the model is for GP without auditory pathology or malformation. Healing could be different depending on the abnormalities causing the deafness, especially in case of pressure abnormality in the inner ear (malformation or hydrops), which could lead to active leakage of perilymph through the perforation.
Line 268: The notion of "permanent hearing damage" is challenged by recent discoveries of covert deafness. Authentic cochlear lesions (synaptopathies) may exist in the aftermath of a cochlear lesion, associated only with a temporary elevation of thresholds. This observation does not rule out a procedure-related cochlear lesion, and this point could be mentioned.
Author Response
Comment: Line 68: the term "elegant" seems too subjective to me and could be replaced by a more neutral term
Response: We agree with this assessment and have replaced the term “elegant” with “novel.” See line 69: “…is a novel means of achieving intracochlear access…”
Comment: Line 69: there is a certain paradox in this sentence, since a perforation is necessarily a form of anatomical damage. I think the wording could be less categorical, such as "with minimal anatomic or functional damage".
Response: We agree with this assessment and have changed the wording as suggested by the reviewer. See line 70: “…with minimal anatomic and functional damage.”
Comment: Line 142- Line 185-Figure 3: The statement that the fibers are spread without being cut seems to me to be difficult to affirm or deny. This is a microscopic observation in which a perforation is observed, the state of the fibers is difficult to analyze. This statement could be nuanced.
Response: We agree that this statement is difficult to affirm or deny. We have deleted our statement in line 142: “As the needle further penetrates the RWM, the taper should push the fibers aside rather than cut the fibers.” We have also made the following changes: Line 184: “…Figure 3b shows collagen and connective fiber separation rather than scission.” Line 222: “…Thus, microneedles with an ultra-sharp tip radius Rt≈1 µm, allowing for fiber separation and not scission, is an achievable design…”
Comment: Lines 194 to 203: the data show complete healing in one week. I think it should be remembered here that the model is for GP without auditory pathology or malformation. Healing could be different depending on the abnormalities causing the deafness, especially in case of pressure abnormality in the inner ear (malformation or hydrops), which could lead to active leakage of perilymph through the perforation.
Response: We agree with this statement. We have added the below sentence: Lines 201-204: “Of note, these experiments were performed in healthy GPs; models of auditory pathology, especially those producing pressure abnormalities in the inner ear, may result in different RWM healing properties.”
Comment: Line 268: The notion of "permanent hearing damage" is challenged by recent discoveries of covert deafness. Authentic cochlear lesions (synaptopathies) may exist in the aftermath of a cochlear lesion, associated only with a temporary elevation of thresholds. This observation does not rule out a procedure-related cochlear lesion, and this point could be mentioned.
Response: We agree that denying permanent hearing loss is problematic. We have modified the statement, and added a statement earlier in the manuscript. Lines 200-201: “…there were no measurable audiologic consequences, though covert hearing loss cannot be ruled out.” Lines 269-270: “There were no measurable shifts in hearing thresholds…”
Reviewer 3 Report
This is a well-written manuscript giving the reader some thoughtful information about an area of hearing research that is critical. Delivery of drugs to the ear has been a major impediment to treatment progress. This is an excellent next step.
Author Response
We appreciate your review and thank you for your positive comments.